

# Hydrogen sulfide protects against cisplatin-induced experimental nephrotoxicity in animal models: a systematic review and meta-analysis

Zhenyuan Han[1,2,*], Tianyu Deng[1,2,*], Dechao Yan[1,2], Yutao Jia[1,2], Jing Tang[3,4] and Xiaoyan Wang[1,2]

[1] Department of Nephrology, The Affiliated BenQ Hospital of Nanjing Medical University, Nanjing Medical University, Nanjing, Jiangsu, China

[2] The Core Laboratory for Clinical Research, Nanjing BenQ Medical Center, Nanjing Medical University, Nanjing, China

[3] Collaborative Innovation Center for Cancer Personalized Medicine, Nanjing Medical University, Nanjing, China

[4] Department of Hematology, The First Affiliated Hospital of Nanjing Medical University, Nanjing Medical University, Nanjing, China

* These authors contributed equally to this work.

Corresponding authors
Jing Tang, 437223501@qq.com
Xiaoyan Wang,
Xiaoyan.Wang@benqmedicalcenter.com

## ABSTRACT

**Background:** Cisplatin-induced acute kidney injury (cis-AKI) is not rare in oncological patients clinically, but there are limited prevention and treatment methods available. The efficacy of hydrogen sulfide ($H_2S$) in mitigating cis-AKI has been studied and determined in animal models.

**Methods:** According to the pre-registered program (PROSPERO: CRD 42023463779), we searched PubMed/Medline, Embase, and Web of Science databases using the keywords: hydrogen sulfide, cisplatin, acute kidney injury, and alternatives. A total of 13 articles met the inclusion criteria were included. Standardized mean difference (SMD) and 95% confidence interval (CI) were calculated and aggregated using random effects meta-analysis.

**Results:** The results showed that $H_2S$ treatment significantly improved renal function (serum creatinine SMD = −2.96, 95% CI [−3.72 to −2.19], $p < 0.00001$; blood urea nitrogen SMD = −2.73, 95% CI [−3.68 to −1.78], $p < 0.00001$), decreased oxidative stress (superoxide dismutase SMD = 2.90, 95% CI [1.36–4.43], $p = 0.0002$) and inflammation levels (interleukin-1β SMD = −4.41, 95% CI [−5.84 to −2.97], $p < 0.00001$). However, there was a high degree of heterogeneity between studies ($I^2 > 70\%$). Further subgroup analysis did not show a clear source of the heterogeneity, but various $H_2S$ donors exhibited positive renal protection in those studies.

**Conclusions:** $H_2S$ could be a new approach for treating cis-AKI, while the differential efficacies among natural and slow-release $H_2S$ donors remain to be compared and evaluated further. This meta-analysis may shed light on establishing preclinical and clinical investigation guidelines for treating human cis-AKI with $H_2S$ donors.

## INTRODUCTION

Cisplatin, as an efficient chemotherapy drug, induces DNA crosslinks, causes DNA damage, and ultimately exerts its anti-cancer effect (*Ghosh, 2019*). Since its discovery in the late 1960s, cisplatin and its derivatives have been widely used in the treatment of various solid tumors, including lung cancer (*Mascaux et al., 2000*), mammary cancer (*Rodler et al., 2023*), bladder cancer (*Kamat et al., 2016a*), and ovarian cancer (*Markman, 1994*). However, cisplatin can cause multiple organ toxicity in the process of tumor treatment, among which kidney toxicity is the most common (*Ries & Klastersky, 1986*). Despite the clinical use of pre-treatment hydration therapy to reduce multi-organ toxicity caused by cisplatin, acute kidney injury (AKI) still occurs in approximately 30% of patients (*Latcha et al., 2016*). The nephrotoxic effects of cisplatin are cumulative and dose-dependent, mainly through the transport of organic cationic transporter-2 and copper transporter-1 in the basolateral membrane of the proximal tubule to intracellular accumulation (*Guo et al., 2018*; *Sharaf, El Morsy & El-Sayed, 2023*), resulting in proximal tubular injury, oxidative stress, inflammation, DNA damage, mitochondrial dysfunction, and vascular injury (*Pabla et al., 2009*; *Yan et al., 2016*), and ultimately AKI. Once AKI occurs and is not under control, the disease will progress into chronic kidney disease (CKD) eventually (*Jiang et al., 2020*; *Xiao et al., 2016*; *Zhang et al., 2023a*). At present, there are no effective drugs or methods to treat kidney damage caused by cisplatin. Therefore, seeking a novel strategy has become a hot topic in the prevention and treatment of cisplatin-induced AKI (cis-AKI).

In the last twenty years, extensive research has been conducted on hydrogen sulfide ($H_2S$) due to its significant functions in preserving cellular membrane stability, controlling cellular differentiation and growth, safeguarding the integrity of mitochondrial DNA, engaging in signal transmission processes, and governing programmed cell death (*Cuevasanta et al., 2012*; *Kamat et al., 2016b*; *Zhang et al., 2019*; *Zhong et al., 2010*). This makes it the third naturally occurring gas molecule, following carbon monoxide and nitric oxide. Evidence has shown that endogenous $H_2S$ is closely involved in the physiological functions and pathological changes of the kidney (*Ahmad et al., 2012*; *Yuan et al., 2019*). According to animal experimental research reports, increasing the amount of $H_2S$ with exogenous $H_2S$ donors can alleviate the renal toxicity caused by cisplatin (*Ahangarpour et al., 2014*; *Cao et al., 2018a*), mainly through anti-inflammatory, antioxidant, autophagy, and other mechanisms (*Ahangarpour et al., 2014*; *Sun et al., 2020*; *Yuan et al., 2019*).

Although $H_2S$ has been proven to be protective against cisplatin-induced nephrotoxicity in animal studies, there is no approved $H_2S$ medication available for patients yet. Thus, conducting meta-analyses on published animal studies to validate $H_2S$ efficacy in cis-AKI is of significance for future clinical investigations. To shed light on the future applications of $H_2S$ in oncology patients with cis-AKI, we reviewed and summarized current

publications and performed a meta-analysis on the current findings about the protective effect of $H_2S$ against nephrotoxicity caused by cisplatin in animal models through *in vivo* studies.

## MATERIALS AND METHODS

### Protocol and registration

The methods used in this study are previously specified in PROSPERO (CRD 42023463779). The systematic review and meta-analysis were conducted according to the guidelines for systematic review and meta-analysis of animal studies and reported under the Preferred Reporting Items for Systematic Reviews and Meta-Analyses (PRISMA) for the manuscript (*Moher et al., 2009*; *Page et al., 2021*).

### Search strategy and study selection

This meta-analysis involved an extensive review of the effects of $H_2S$ therapy on animal models of cis-AKI. All relevant original research articles published up to March 2024 were included in the review. We systematically searched through Web of Science, Scopus, and Pub-Med/Medline by using the following keywords: "hydrogen sulfide [Mesh], cisplatin [Mesh], acute kidney injury [Mesh]", and alternatives.

Two independent reviewers (Han and Yan) selected literature that met the inclusion criteria by screening titles and abstracts. According to the inclusion and exclusion criteria (shown below), the article was classified as irrelevant and relevant. Any existing disagreements were resolved through discussion with a third reviewer (Deng), utilizing the identical approach separately.

The full text of qualified literature was searched as far as possible. In case the full text was not available, we applied to the first author of the original article by email. Two authors examined the full text to exclude studies that did not meet predetermined inclusion criteria. In addition, we carefully examined the references of the selected literature to identify those that met the research selection criteria. The sorting process is managed using the bibliographic management software Endnote 20.

### Inclusion and exclusion criteria

We established the following criteria for inclusion in this review study: (1) literature published in English, (2) the experiments were carried out in animals, (3) wild-type animals without any genetic or environmental modifications, (4) experimental cis-AKI model of kidney, (5) $H_2S$ donors were given in any route, frequency, and dose to treat/prevent cis-AKI. If the following conditions exist, the literature was excluded from this meta-analysis: (1) any *in vitro*, and computer modeling studies, (2) studies conducted on genetically modified animals or animals with complications, such as diabetes mellitus, myocardial infarction, heart failure, or hypertension, (3) research conducted in humans, (4) studies on simultaneous administration of $H_2S$ and other drugs, (5) reviews and conference reports.

## Data extraction

The experimental data presented in each article were extracted and analyzed: experimental design; animal characteristics of species, gender, age, or weight; methods of administering cisplatin including mode, dosage, and duration; $H_2S$ donor administration details of type, dosage, duration, and route. The main outcome variable was the value of serum creatinine. Secondary outcome variables were markers of kidney injury, inflammation, oxidative stress, and cell apoptosis. Histopathological images were independently evaluated by two reviewers using a semi-quantitative scoring system based on HE/PAS staining. Tubular injury was graded as follows: 0 (no injury), 1 (<25% of tubules affected), 2 (25–50%), 3 (50–75%), and 4 (>75%), consistent with established criteria (*Gong et al., 2021*; *Yuan et al., 2019*). Discrepancies were resolved by consensus or third-party arbitration. The specific plan could be found in the PROSPERO database (CRD 42023405958). The data extraction from graphs was facilitated by employing digital software (WebPlotDigitizer, Version 4.6, https://automeris.io/, September 2022), which was developed by Ankit Rohatgi in Pacifica, California, USA. The data collection was conducted independently by two authors (Deng and Yan), ensuring unbiased and reliable results. If the difference between the data extracted by the two examiners was <10%, the average of the two values was taken. Any discrepancies were resolved through discussion with a third reviewer (Han), applying the same technique independently, to ensure consensus and accuracy.

## Quality assessment

Risk of bias and quality of evidence from included studies were assessed according to SYR-CLE's Risk of Bias tool (*Hooijmans et al., 2014*), which had been adapted for animal experimental models. Two reviewers conducted separate evaluations of bias risk and study quality for each study, with any discrepancies being resolved through consultation with a third reviewer. The risk-of-bias plots were generated by using Microsoft Excel (2312 Build 16.0.17126.20132; Microsoft, Bellingham, WA, USA) and an online application, robvis (accessible at https://mcguinlu.shinyapps.io/robvis/).

## Statistical analysis

Data were presented as SMD of 95% confidence intervals (CI). Standard deviation (STD) was utilized to compare the $H_2S$ treatment group with the control group (no treatment/placebo). The level of heterogeneity was measured using $I^2$, Tau2, and Q statistical tests. $I^2$ value exceeding 50% indicated significant heterogeneity. Forest plots were generated to summarize the findings from meta-analysis studies. Meta-regression analyses and subgroup analyses were conducted to explore associations among animal species, type of $H_2S$ donor, and route of administration in relation to outcome variables. Publication bias was evaluated through examination of funnel plots and quantified using the Egger test. A *p*-value less than 0.05 was considered statistically significant. Review Manager (version RevMan 5.3) and Stata (StataCorp, College, TX, USA) were utilized for statistical analysis and figure creation.
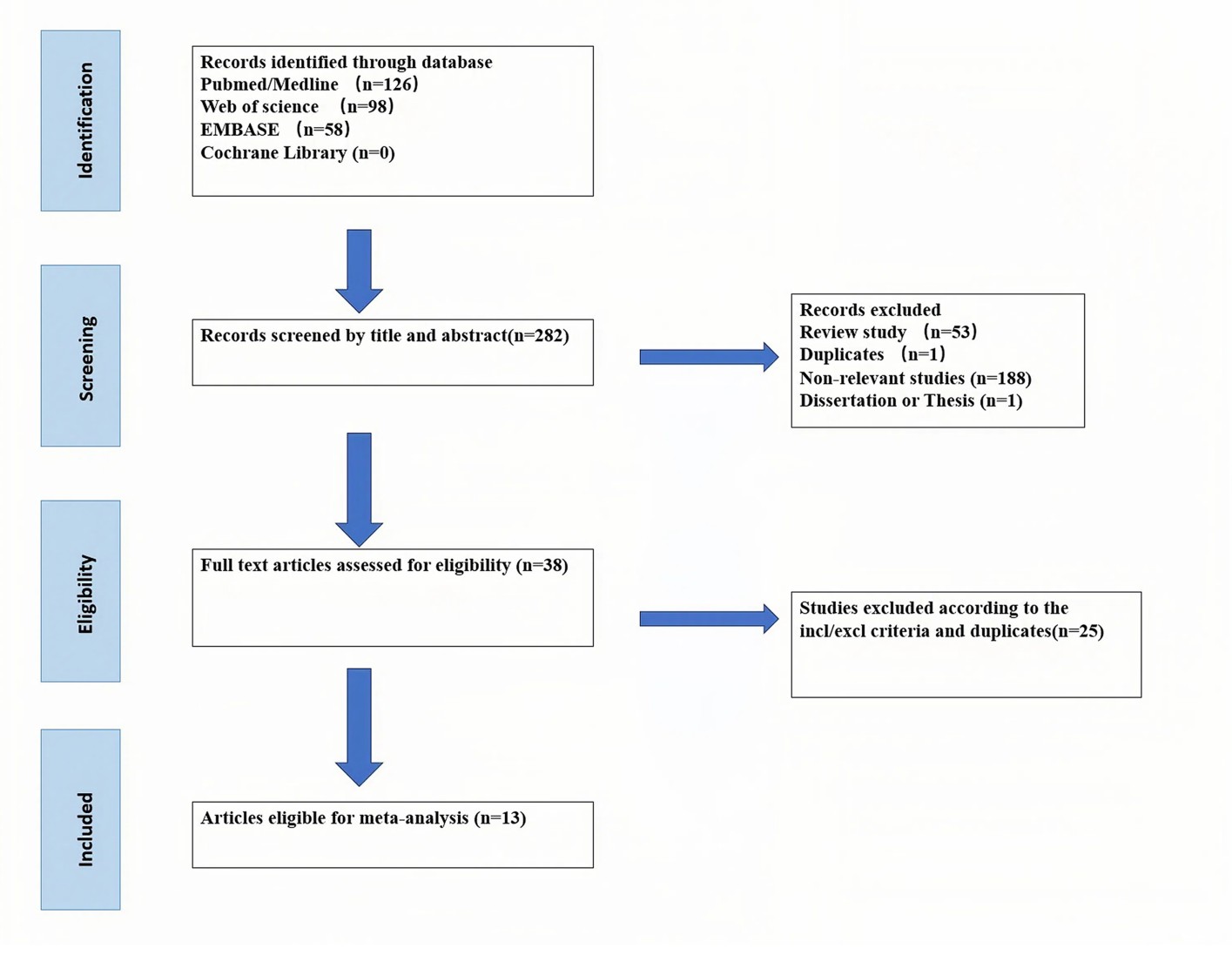

**Figure 1 PRISMA flowchart of the search and selection process of the article.**

# RESULTS

## Study selection process

The process of retrieving literature was illustrated in Fig. 1. A total of 282 articles were initially gathered from PubMed/MEDLINE, Web of Science, Cochrane Library, and Scopus databases. Following the removal of duplicates and exclusion criteria, a meta-analysis was conducted on 13 literature (*Abdel-Daim et al., 2019*; *Ahangarpour et al., 2014*; *Ahmad, 2022*; *Cai et al., 2024*; *Cao et al., 2018a, 2018b*; *Chiarandini Fiore et al., 2008*; *Elkhoely & Kamel, 2018*; *Jiang et al., 2024*; *Karimi et al., 2017*; *Sun et al., 2020*; *Wang, Liu & Liu, 2022*; *Yuan et al., 2019*). The characteristics and experimental protocols of the included studies were listed in Table 1. Eight exogenous $H_2S$ donors were administered to elevate systemic $H_2S$ levels in animal models. These included synthetic compounds

**Table 1 Main characteristics of included studies.**

| Reference | Species | Dosage (cisplatin, i.p) | Treatment duration (day) (cispaltin) | Administration (H$_2$S) | Donor | Dosage (H$_2$S) | Treatment duration (day) (H$_2$S) | Sample (cis vs cis +H$_2$S) |
|---|---|---|---|---|---|---|---|---|
| Cai et al. (2024) | C57BL/6 mice | 25 mg/kg | 4 | po | ADT-OH | 37 mg/kg | 4 | 6 vs 6 |
| Cai et al. (2024) | C57BL/6 mice | 25 mg/kg | 4 | po | GYY4137 | 100 mg/kg | 4 | 6 vs 6 |
| Cai et al. (2024) | C57BL/6 mice | 25 mg/kg | 4 | po | NaHS | 5.6 mg/kg | 4 | 6 vs 6 |
| Jiang et al. (2024) | BALb/c mice | 20 mg/kg | 5 | i.p | DATS | 20 mg/kg | 20 | 8 vs 8 |
| Wang, Liu & Liu (2022) | beagle | 5 mg/kg | 3 | i.v | NaHS | 1 mg/kg/h | 3 | 6 vs 6 |
| Ahmad (2022) | Wistar rat | 5 mg/kg | 7 | i.p | NaHS | 56 µmol/kg | 35 | 6 vs 6 |
| Sun et al. (2020) | C57BL/6 mice | 25 mg/kg | 3 | i.p | NaHS | 5.6 mg/kg | 4 | 5 vs 5 |
| Sun et al. (2020) | C57BL/6 mice | 25 mg/kg | 3 | i.p | GYY4137 | 100 mg/kg | 4 | 5 vs 5 |
| Sun et al. (2020) | C57BL/6 mice | 25 mg/kg | 3 | i.p | Na$_2$S$_4$ | 500 µg/kg | 4 | 5 vs 5 |
| Yuan et al. (2019) | C57BL/6 mice | 16 mg/kg | 4 | i.p | NaHS | 5.6 mg/kg | 4 | 8 vs 8 |
| Abdel-Daim et al. (2019) | Wistar rat | 7 mg/kg | 7 | po | Allicin | 10 mg/kg | 14 | 8 vs 8 |
| Elkhoely & Kamel (2018) | SD rat | 3.5 mg/kg | 4 | i.g | DAS | 50 mg/kg | 4 | 6 vs 6 |
| Elkhoely & Kamel (2018) | SD rat | 3.5 mg/kg | 4 | i.g | DAS | 100 mg/kg | 4 | 6 vs 6 |
| Cao et al. (2018a, 2018b) | SD rat | 7 mg/kg | 6 | i.p | Na$_2$S$_4$ | 5.6 mg/kg | 7 | 8 vs 8 |
| Cao et al. (2018a) | C57BL/6 mice | 25 mg.kg | 3 | i.p | NaHS | 5.6 mg/kg | 4 | 9 vs 9 |
| Cao et al. (2018b) | C57BL/6 mice | 25 mg.kg | 3 | i.p | GYY4137 | 100 mg/kg | 4 | 9 vs 9 |
| Karimi et al. (2017) | SD rat | 5 mg/kg | | i.p | NaHS | 200 µg/kg | 15 | 8 vs 8 |
| Ahangarpour et al. (2014) | SD rat | 6 mg/kg | 6 | i.p | NaHS | 100 µmol/kg | 6 | 10 vs 10 |
| Chiarandini Fiore et al. (2008) | SD rat | 10.5 mg/kg | 4 | i.g | DADS | 292.5 mg/kg (1 day) 146.25 mg/kg (3 days) | 4 | 8 vs 8 |

Note:
(DADS: diallyl disulfide, DAS: diallyl sulfide, DATS: diallyl trisulfide, ip: intraperitoneal injection, ig: intragastric, iv: intravenous injection, po: peros(Latin)- oral administration).
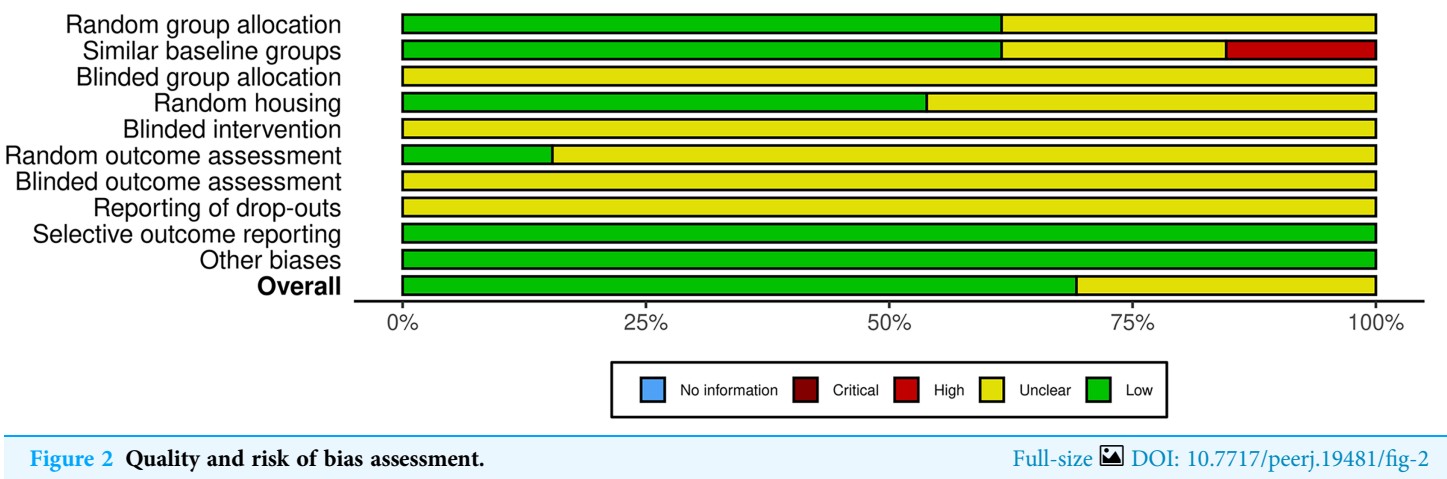

**Figure 2 Quality and risk of bias assessment.**

(ADT-OH, NaHS, $Na_2S_4$, GYY4137) and natural plant extracts (Allicin, diallyl disulfide (DADS), diallyl sulfide (DAS), diallyl trisulfide (DATS)), which either directly release $H_2S$ or were metabolized into $H_2S$ *in vivo*. Four administration routes were used: intra-peritoneal injection, intravenous injection, oral administration, and gavage. Certainly, different donor drugs and animal species require adjustment and optimization of the corresponding drug dose.

## Risk of bias assessment

Using the SYRCLE bias risk tool to evaluate the quality of animal studies revealed that all studies had unknown bias in the blind design of grouping, intervention, outcome assessment, inclusion, and exclusion of reports. Among the 13 included pieces of literature, nine had low bias, four had unknown bias, and no literature had high bias (Fig. 2 and Fig. S1).

## Renal function recovery after $H_2S$ treatment on cis-AKI

Serum creatinine and urea nitrogen were measured in 19 studies. Renal function was significantly improved after $H_2S$ donors postprocessing, with creatinine (SMD = −2.96, 95% CI [−3.72, −2.19], Z = 7.54, $p < 0.00001$; Fig. 3) and urea nitrogen (SMD = −2.73, 95% CI [−3.68, −1.78], Z = 5.62, $p < 0.00001$; Fig. 4). The overall effect size was accompanied by a high degree of heterogeneity ($I^2 = 73\%$, $p < 0.00001$, and $I^2 = 75\%$, $p < 0.00001$, respectively).

## Effects of $H_2S$ on oxidative stress, inflammation and apoptosis

Superoxide dismutase (SOD) was measured in six studies. A meta-analysis of these studies showed that $H_2S$ treatment was associated with significantly increased SOD levels (SMD = 2.90, 95% CI [1.36–4.43], Z = 3.70, $p = 0.0002$; Fig. 5), the heterogeneity was statistically significant ($I^2 = 79\%$, $p = 0.0001$). Additionally, nine separate research inquiries examining malondialdehyde (MDA) concentrations demonstrated that subjects who underwent $H_2S$ adaptation exhibited considerably reduced MDA levels when compared to those who received no treatment (SMD = −2.85, 95% CI [−3.98 to −1.71], Z = 4.91,

| Study or Subgroup | H2S+Cis Mean | SD | Total | Cis Mean | SD | Total | Weight | Std. Mean Difference IV, Random, 95% CI | Std. Mean Difference IV, Random, 95% CI |
|---|---|---|---|---|---|---|---|---|---|
| Abdel-Daim et al.2019 | 1.2 | 0.281421 | 8 | 3.33 | 0.567843 | 8 | 5.2% | -4.49 [-6.54, -2.45] | |
| Ahangarpour et al.2014 | 0.59 | 0.159057 | 10 | 1.25 | 0.282302 | 10 | 6.7% | -2.76 [-4.05, -1.46] | |
| Ahmad et al.2022 | 1.82 | 0.08 | 6 | 2.21 | 0.13 | 6 | 5.3% | -3.34 [-5.32, -1.35] | |
| Cai et al.2024a | 0.810145 | 0.38328 | 6 | 4.571174 | 0.707267 | 6 | 3.4% | -6.10 [-9.29, -2.92] | |
| Cai et al.2024b | 2.116447 | 0.647975 | 6 | 4.571174 | 0.707267 | 6 | 5.3% | -3.34 [-5.33, -1.36] | |
| Cai et al.2024c | 1.369888 | 0.285048 | 6 | 4.571174 | 0.707267 | 6 | 3.8% | -5.48 [-8.39, -2.58] | |
| Cao et al.2018 | 107.5337 | 22.90105 | 8 | 151.1351 | 48.29226 | 8 | 7.1% | -1.09 [-2.16, -0.02] | |
| Cao et al.2018a | 126.3747 | 20.30631 | 9 | 164.602 | 46.52115 | 9 | 7.2% | -1.01 [-2.01, -0.02] | |
| Cao et al.2018b | 112.5125 | 23.50033 | 9 | 164.602 | 46.52115 | 9 | 7.2% | -1.35 [-2.40, -0.30] | |
| Elkhoely et al.2018a | 3.02 | 0.19 | 6 | 6.11 | 0.29 | 6 | 1.5% | -11.63 [-17.43, -5.84] | |
| Elkhoely et al.2018b | 1.82 | 0.17 | 6 | 6.11 | 0.29 | 6 | 0.8% | -16.66 [-24.87, -8.45] | |
| Fiore et al.2008 | 14.83 | 2.07 | 8 | 24.74 | 3.03 | 8 | 5.8% | -3.61 [-5.35, -1.87] | |
| Jiang et al.2023 | 33.49789 | 3.284131 | 8 | 65.96848 | 4.247367 | 8 | 3.2% | -8.09 [-11.46, -4.71] | |
| Karimi et al.2017 | 0.6 | 0.1 | 8 | 0.8 | 0.11 | 8 | 6.8% | -1.80 [-3.01, -0.58] | |
| Sun et al.2020a | 63.34206 | 26.46039 | 5 | 120.3299 | 27.24807 | 5 | 6.0% | -1.92 [-3.56, -0.27] | |
| Sun et al.2020b | 61.41796 | 22.16575 | 5 | 120.3299 | 27.24807 | 5 | 5.8% | -2.14 [-3.87, -0.41] | |
| Sun et al.2020c | 69.80869 | 19.94439 | 5 | 120.3299 | 27.24807 | 5 | 6.0% | -1.91 [-3.55, -0.27] | |
| Wang et al.2022 | 212.2239 | 28.99805 | 6 | 300.7694 | 47.75598 | 6 | 6.2% | -2.07 [-3.59, -0.55] | |
| Yuan et al.2019 | 31.16556 | 10.60318 | 8 | 62.38402 | 15.52657 | 8 | 6.6% | -2.22 [-3.54, -0.90] | |
| **Total (95% CI)** | | | 133 | | | 133 | 100.0% | -2.96 [-3.73, -2.19] | |

Heterogeneity: Tau² = 1.86; Chi² = 67.82, df = 18 (P < 0.00001); I² = 73%
Test for overall effect: Z = 7.54 (P < 0.00001)

**Figure 3 Experimental cisplatin *vs*. H₂S forest plot showing the pooled effect of H₂S treatment on serum creatinine level.** Note: *Abdel-Daim et al. (2019), Ahangarpour et al. (2014), Ahmad (2022), Cai et al. (2024), Cao et al. (2018a, 2018b), Chiarandini Fiore et al. (2008), Elkhoely & Kamel (2018), Jiang et al. (2024), Karimi et al. (2017), Sun et al. (2020), Wang, Liu & Liu (2022), Yuan et al. (2019).*

| Study or Subgroup | H2S+Cis Mean | SD | Total | Cis Mean | SD | Total | Weight | Std. Mean Difference IV, Random, 95% CI | Std. Mean Difference IV, Random, 95% CI |
|---|---|---|---|---|---|---|---|---|---|
| Cai et al.2024a | 1.18 | 0.3 | 6 | 5.19 | 0.66 | 6 | 4.3% | -7.22 [-10.92, -3.52] | |
| Cai et al.2024b | 2.02 | 0.38 | 6 | 5.19 | 0.66 | 6 | 5.8% | -5.43 [-8.32, -2.55] | |
| Cai et al.2024c | 1.76 | 0.52 | 6 | 5.19 | 0.66 | 6 | 5.9% | -5.33 [-8.17, -2.49] | |
| Cao et al.2018 | 205.08 | 52.36 | 8 | 270.61 | 111.26 | 8 | 11.0% | -0.71 [-1.73, 0.31] | |
| Cao et al.2018a | 141.62 | 33.55 | 9 | 196.7 | 58.83 | 9 | 11.0% | -1.10 [-2.10, -0.09] | |
| Cao et al.2018b | 117.53 | 25.66 | 9 | 196.7 | 58.83 | 9 | 10.7% | -1.66 [-2.77, -0.55] | |
| Jiang et al.2023 | 24.25 | 3.2 | 8 | 50.1 | 4.27 | 8 | 6.1% | -6.48 [-9.24, -3.71] | |
| Sun et al.2020a | 17.97 | 11.31 | 5 | 48.28 | 22.54 | 5 | 9.5% | -1.54 [-3.05, -0.02] | |
| Sun et al.2020b | 16.87 | 5.43 | 5 | 48.28 | 22.54 | 5 | 9.3% | -1.73 [-3.31, -0.15] | |
| Sun et al.2020c | 21.18 | 5.93 | 5 | 48.28 | 22.54 | 5 | 9.6% | -1.49 [-2.98, 0.01] | |
| Wang et al.2022 | 7.28 | 1 | 6 | 17.62 | 2.92 | 6 | 6.9% | -4.37 [-6.79, -1.96] | |
| Yuan et al.2019 | 41.47 | 13.57 | 8 | 87.06 | 18.78 | 8 | 9.8% | -2.63 [-4.07, -1.19] | |
| **Total (95% CI)** | | | 81 | | | 81 | 100.0% | -2.73 [-3.68, -1.78] | |

Heterogeneity: Tau² = 1.87; Chi² = 43.25, df = 11 (P < 0.00001); I² = 75%
Test for overall effect: Z = 5.62 (P < 0.00001)

**Figure 4 Experimental cisplatin *vs*. H₂S forest plot showing the pooled effect of H₂S treatment on BUN level.** Note: *Cai et al. (2024), Cao et al. (2018a, 2018b), Jiang et al. (2024), Sun et al. (2020), Wang, Liu & Liu (2022), Yuan et al. (2019).* 

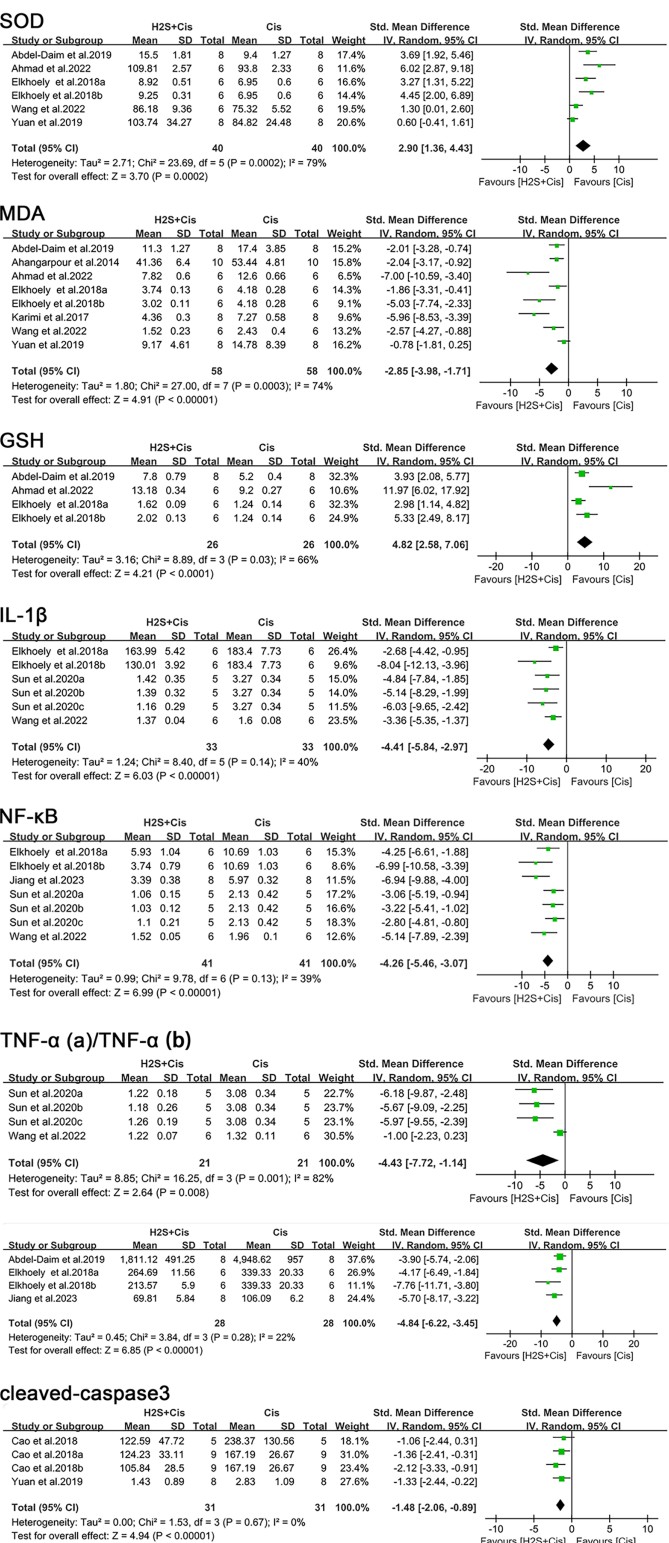

**Figure 5 Experimental cisplatin *vs*. H₂S forest plot showing the pooled effect of H2S treatment on SOD, MDA, GSH, IL- 1β, NF-κ B, TNF- α, and cleaved-caspase3 levels.** Note: *Abdel-Daim et al. (2019)*, *Ahangarpour et al. (2014)*, *Ahmad (2022)*, *Cai et al. (2024)*, *Cao et al. (2018a, 2018b)*, *Elkhoely & Kamel (2018)*, *Jiang et al. (2024)*, *Karimi et al. (2017)*, *Sun et al. (2020)*, *Wang, Liu & Liu (2022)*, *Yuan et al. (2019)*.

| Study or Subgroup | H2S+Cis Mean | SD | Total | Cis Mean | SD | Total | Weight | Std. Mean Difference IV, Random, 95% CI | Std. Mean Difference IV, Random, 95% CI |
|---|---|---|---|---|---|---|---|---|---|
| Ahangarpour et al.2014 | 2.3 | 0.13 | 10 | 4.75 | 0.12 | 10 | 4.1% | -18.76 [-25.30, -12.21] | |
| Ahmad et al.2022 | 1.4 | 0.21 | 6 | 4.45 | 0.22 | 6 | 4.2% | -13.09 [-19.58, -6.60] | |
| Cai et al.2024a | 2.610606 | 0.13 | 6 | 3.194697 | 0.32 | 6 | 8.1% | -2.21 [-3.77, -0.64] | |
| Cai et al.2024b | 2.129545 | 0.14 | 6 | 3.194697 | 0.34 | 6 | 7.7% | -3.78 [-5.95, -1.62] | |
| Cai et al.2024c | 1.515152 | 0.12 | 6 | 3.194697 | 0.2 | 6 | 5.5% | -9.40 [-14.13, -4.67] | |
| Cao et al.2018 | 2.05 | 0.111 | 8 | 4.75 | 0.18 | 8 | 3.9% | -17.07 [-23.95, -10.19] | |
| Elkhoely et al.2018a | 2.45 | 0.21 | 6 | 3.65 | 0.22 | 6 | 7.2% | -5.15 [-7.91, -2.39] | |
| Elkhoely et al.2018b | 1.25 | 0.2 | 6 | 3.65 | 0.14 | 6 | 4.3% | -12.83 [-19.20, -6.47] | |
| Fiore et al.2008 | 2.25 | 0.15 | 8 | 4.35 | 0.19 | 8 | 5.5% | -11.60 [-16.33, -6.87] | |
| Jiang et al.2023 | 2.15 | 0.165 | 8 | 3.45 | 0.18 | 8 | 7.0% | -7.12 [-10.12, -4.11] | |
| Karimi et al.2017 | 1.4 | 0.16 | 8 | 4.45 | 0.43 | 8 | 6.4% | -8.89 [-12.57, -5.21] | |
| Sun et al.2020a | 3.4 | 0.42 | 5 | 4.4 | 0.21 | 5 | 7.8% | -2.72 [-4.69, -0.75] | |
| Sun et al.2020b | 3.2 | 0.4 | 5 | 4.4 | 0.233 | 5 | 7.6% | -3.31 [-5.55, -1.07] | |
| Sun et al.2020c | 3.1 | 0.44 | 5 | 4.4 | 0.21 | 5 | 7.6% | -3.41 [-5.69, -1.12] | |
| Wang et al.2022 | 2.55 | 0.33 | 6 | 4.35 | 0.13 | 6 | 6.6% | -6.62 [-10.05, -3.20] | |
| Yuan et al.2019 | 1.25 | 0.32 | 8 | 3.95 | 0.31 | 8 | 6.7% | -8.10 [-11.48, -4.72] | |
| **Total (95% CI)** | | | **107** | | | **107** | **100.0%** | **-7.29 [-9.14, -5.44]** | |

Heterogeneity: Tau² = 10.42; Chi² = 84.27, df = 15 (P < 0.00001); I² = 82%
Test for overall effect: Z = 7.72 (P < 0.00001)

**Figure 6 Experimental cisplatin *vs*. H$_2$S forest plot showing the pooled effect of H$_2$S treatment on renal tubule injury.** Note: *Ahangarpour et al. (2014), Ahmad (2022), Cai et al. (2024), Cao et al. (2018a), Chiarandini Fiore et al. (2008), Elkhoely & Kamel (2018), Jiang et al. (2024), Karimi et al. (2017), Sun et al. (2020), Wang, Liu & Liu (2022), Yuan et al. (2019).*

$p < 0.00001$; Fig. 5), with considerable variation observed across all included studies ($I^2 = 74\%$, $p = 0.0003$). As for glutathione (GSH), statistically reported by four studies, as well as SOD activity, treatment of H$_2$S induced a significant elevation (SMD = 4.82, 95% CI [2.58–7.06], Z = 4.21, $p < 0.0001$; $I^2 = 66\%$, $p = 0.03$, Fig. 5).

In contrast, cotreatment of cisplatin with H$_2$S exhibited a significant decrease of these pro-inflammatory markers compared to the cis group as follows: IL-1β (SMD = −4.41, 95% CI [−5.84 to −2.97], Z = 6.03, $p < 0.00001$), NF-κB (SMD = −4.26, 95% CI [−5.46 to −3.07], Z = 6.99, $p < 0.0001$), TNF-α (SMD = −4.43, 95% CI [−7.72 to −1.14], Z = 2.64, $p = 0.008$, Fig. 5; SMD = −4.84, 95% CI [−6.22 to −3.45], Z = 6.85, $p < 0.00001$, Fig. 5).

H$_2$S treatment also reduced apoptosis and tissue damage, such as a notable decrease in cleaved-caspase-3 level (SMD = −1.48, 95% CI [−2.06 to −0.89], Z = 4.94, $p < 0.00001$, Fig. 5) and renal tubule injury (SMD = −7.29, 95% CI [−9.14 to −5.44], Z = 7.72, $p < 0.00001$, Fig. 6). Unfortunately, a suitable meta-analysis could not be conducted due to the lack of quantitative data available for the TUNEL-positive cells.

### Sensitivity analysis

Aggregating SMDS with fixed-effects models instead of random-effects models resulted in consistently large and significant effect estimates, with no significant change in the heterogeneity of outcome measures. Sensitivity analysis confirmed the consistency of the findings even after excluding individual studies (Fig. S2).

## Heterogeneity analysis

To examine possible factors contributing to heterogeneity, we conducted subgroup analyses on the overall effect of, $H_2S$ donor type, animal species, and administration method on serum creatinine concentration. Each study showed a protective effect (SMD = −2.96, 95% CI [−3.72 to −2.19], $I^2$ = 73%, $p < 0.00001$; SMD = −2.96, 95% CI [−3.72 to −2.19], $I^2$ = 73%, $p < 0.00001$; SMD = −2.96, 95% CI [−3.72 to −2.19], $I^2$ = 73%, $p < 0.00001$, Figs. S3–S5). However, compared with the results before grouping, there was no significant reduction in heterogeneity, and the intergroup heterogeneity was large ($I^2$ = 72.4%, $p = 0.006$; $I^2$ = 0%, $p = 0.47$; $I^2$ = 81.3%, $p = 0.001$, Figs. 3–5). Although the difference between groups was low in the species subgroup analysis, it was not statistically significant.

## Publication bias

Visually, the funnel plot revealed potential asymmetry (Fig. S6). Two studies were clearly outside the 95% confidence interval, indicating publication bias, which was supported by the Egger regression test (Intercept = −5.08, 95% CI [−5.88 to −4.28], $p = 0.000$, Fig. S7). After the application of the clipping method, the number of additions was 0, indicating that despite the bias, it had little impact on the conclusion and the results were stable (SMD = −3.265, 95% CI [−4.095 to −2.435], $p = 0.000$, Fig. S8).

## DISCUSSION

Currently, this is the first attempt to conduct a meta-analysis on $H_2S$ efficacy for cisplatin-induced experimental nephrotoxicity. This meta-analysis assessed the consistency of results in animal studies on using $H_2S$ for treating cis-AKI. According to all 13 articles reviewed and analyzed, $H_2S$ donors effectively mitigated renal toxicity induced by cisplatin and significantly re-stored renal function after cisplatin treatment. Therefore, $H_2S$ could be a new approach for treating cis-AKI, theoretically. This meta-analysis may shed light on establishing preclinical and clinical investigation guidelines for treating human cis-AKI with $H_2S$ donors.

Although cis-AKI is not rare in oncological patients clinically, there are limited prevention and treatment methods available, except for adequate hydration (*Launay-Vacher et al., 2008*). According to a meta-analysis on clinical prevention of cisplatin nephrotoxicity, magnesium supplementation could be a forward-thinking and cost-effective preventive strategy (*Hamroun et al., 2019*), which was supported by several animal studies (*Solanki et al., 2014*; *Yokoo et al., 2009*). On the other hand, a meta-analysis conducted by *Emre Aydıngöz et al. (2023)* also shows a protective effect of $H_2S$ on ischemia-reperfusion injury (IRI), also a common AKI model, demonstrating that $H_2S$ effectively prevents or improves renal IRI before, during, or after ischemia and during the reperfusion phases (*Emre Aydıngöz et al., 2023*).

In the renal system, $H_2S$ biosynthesis is primarily regulated by three enzymes: cystathionine-γ-lyase (CSE), cystathionine-β-synthase (CBS), and 3-mercaptopyruvate sulfurtransferase (3-MST) (*Cirino, Szabo & Papapetropoulos, 2023*). These enzymes are abundantly localized in the brush border and cytoplasm of glomerular endothelial cells,

proximal tubules, distal tubules, and peritubular capillary epithelial cells, establishing the kidneys as a critical site for endogenous $H_2S$ production involved in physiological regulation (*Lu et al., 2022*). Subsequent studies identified CBS and CSE as the predominant $H_2S$-synthesizing enzymes in renal tissues, with CBS predominantly expressed in proximal tubular cells of the outer cortex, while CSE exhibited predominant localization in the inner cortex and outer medulla (*Cirino, Szabo & Papapetropoulos, 2023*). In cis-AKI, impairment of endogenous $H_2S$-synthesizing enzymes played a pivotal role in renal pathophysiology. Experimental evidence demonstrated that cisplatin treatment significantly downregulated CSE expression in proximal tubular cells, a phenomenon mechanistically linked to excessive ROS accumulation (*Cao et al., 2018b*; *Jiang et al., 2024*). ROS may directly damage enzyme structures or indirectly suppress transcriptional activity through pro-inflammatory pathways such as NF-κB activation (*Elkhoely & Kamel, 2018*). Animal studies further corroborated that cisplatin administration markedly reduced renal $H_2S$ levels, while pharmacological inhibition of CSE using DL-propargylglycine (PAG) exacerbated tubular necrosis, inflammatory infiltration, and oxidative damage (*Ahmad, 2022*; *Cao et al., 2018b*). These findings underscored the critical disruption of endogenous $H_2S$ biosynthesis in driving cis-AKI progression. Notably, plasma $H_2S$ concentrations also declined during cis-AKI, reflecting systemic depletion of this cytoprotective molecule. The reduction in both renal and circulating $H_2S$ levels correlated with impaired antioxidant defenses, amplified mitochondrial dysfunction, and enhanced susceptibility to cisplatin-induced apoptotic signaling cascades (*Ahmad, 2022*; *Cai et al., 2024*; *Cao et al., 2018b*). The suppression of $H_2S$-synthesizing enzymes served as a central mechanism linking cisplatin toxicity to renal injury.

Our analysis has explored the therapeutic potential and underlying mechanisms of $H_2S$ donors in cisplatin-associated nephrotoxicity, including their ability to inhibit the upregulation of inflammatory factors, reduce ROS production, and alleviate apoptosis and death of tubule cells. NaHS improved mitochondrial energy metabolism by sulfhydrating sirtuin 3 (SIRT3) on both CXXC zinc finger motifs, increasing SIRT3 expression and enhancing its deacetylase activity (*Yuan et al., 2019*). There is also a direct interaction between $H_2S$ and p47phox, which inhibits NADPH oxidase activity and reduces ROS accumulation by persulfide p47phox (*Cao et al., 2018b*). Moreover, by inhibiting the ROS/ mitogen-activated protein kinase signaling pathway, the polysulfide donor $Na_2S_4$ promoted nuclear factor erythroid two-related factor 2 translocation into the nucleus, reduced NOX activation, and alleviated renal toxicity (*Cao et al., 2018a*). Drawing on our previous research, we also found that GYY4137, a long-acting $H_2S$ donor, provides a stable concentration of $H_2S$ in solution for more than 24 h, exerting a strong protective effect on diabetic renal damage *via* affecting multiple ROS-associated enzymes (*Chen et al., 2023*; *Yu et al., 2021*). $Na_2S_4$, NaHS, and GYY4137 similarly reduced the production of inflammatory cytokines and kidney inflammation caused by cisplatin. This mechanism of action is associated with S-sulfhydrylation of signal transduction and transcriptional activator3 (STAT3) and inhibitor kappaB kinaseβ (IKK-β), leading to phosphorylation and decreased expression of pro-inflammatory genes (*Sun et al., 2020*). ADT-OH, a new sustained-release donor of $H_2S$, is found to reduce cisplatin-induced RTEC cell death and

mitochondrial dysfunction in HK-2 cells compared to control cells, indicating its potential for treating cisplatin-induced acute kidney injury (*Cai et al., 2024*). Aside from these potential mechanisms, $H_2S$ also reduces lipid deposition, inhibits ferroptosis, and protects renal tubular cells (*Cai et al., 2024*; *da Costa Marques et al., 2022*; *Zhang et al., 2023b*). These studies also showcased that $H_2S$ donor molecules curtailed inflammation and enhanced mitochondrial function in animal models of cis-AKI.

The high heterogeneity, with $I^2 = 73\%$ on Scr and $I^2 = 75\%$ on BUN, respectively, may be due to the differences in animal species, sample size, and methodology among the studies. In terms of experimental animals, only one study used dogs as experimental subjects (*Wang, Liu & Liu, 2022*), and the rest used rats (eight studies) or mice (10 studies). The lack of significance in animal species subgroup analysis may be attributed to this factor (SMD = −2.96, 95% CI [−3.72 to −2.19], $p = 0.38$, $I^2 = 0\%$, Fig. S4). Due to the lack of data from large mammals, there is still a long way to go for subsequent clinical translation. There were different effect ranges for different $H_2S$ donors seen in our study. This suggested that different donors may have varying degrees of renal protection, and further research is needed to identify the most suitable donor for specific applications. We also observed significant differences in the release curve (*Jiang et al., 2024*), physicochemical and biological characteristics, time of action, and potential therapeutic applications of $H_2S$ donors in animals (*Cai et al., 2024*). These differences highlighted the complexity of $H_2S$ donor research and the need for a deeper understanding of their mechanisms of action. NaHS was the most commonly used donor in eight studies, accounting for 51% of the total. The NaHS group demonstrated the narrowest 95% CI and a low level of heterogeneity in subgroup analyses of donors (95% CI [−2.65 to −1.53], $I^2 = 20\%$, Fig. S3). Moreover, based on the above analysis, GYY4137, used in three studies, stands out among other H2S donors due to its sustained release characteristics. As mentioned above, ADT-OH is more effective than other $H_2S$ donors and offers protection against cisplatin-induced renal tubular cell apoptosis, oxidative stress, and mitochondrial dysfunction *in vitro* (*Cai et al., 2024*). Allicin is the main bioactive compound in garlic, which can be decomposed into DADS, DAS, and DATS (*Bautista et al., 2005*; *Borlinghaus et al., 2014*). *I. vivo*, red blood cells converted these compounds into $H_2S$, producing experimental results similar to those of other artificial $H_2S$ donors (*Benavides et al., 2007*). With the deepening understanding of various garlic extracts, researchers gradually integrate their pharmacological properties, such as anti-inflammatory (*Zhu et al., 2022*), immune regulation (*Arreola et al., 2015*), pathogen infection resistance (*Hall et al., 2017*), antioxidant stress (*Vazquez-Prieto et al., 2011*), and organ protection (*Abdel-Daim et al., 2020*). There were five studies using garlic extracts in this meta-analysis (*Abdel-Daim et al., 2019*; *Chiarandini Fiore et al., 2008*; *Elkhoely & Kamel, 2018*; *Jiang et al., 2024*), and the final results confirmed their inhibitory effect on cisplatin-induced nephrotoxicity (SMD = −7.27, 95% CI [−10.37 to −4.16], $p = 0.0009$, $I^2 = 79\%$, Fig. S3). As the four garlic extracts were only utilized in 1–2 studies, we amalgamated them into a single major category that was applied for initial subgroup analysis, showing that natural $H_2S$ donors possess equivalent therapeutic potential to synthetic donors. To ensure precision, separate subgroup analyses of these four extracts were also conducted, and their renal protective effects remained unchanged despite high

heterogeneity ($I^2 = 85.6\%$, $p < 0.00001$, Fig. S9). However, it is worth noting that there may be some structural limitations on those natural donors in clinical translation, and by-products during the release of $H_2S$ (*El-Saber Batiha et al., 2020*). More accurate extraction and modifications of those compounds remain to be established.

There are common issues in meta-analyses of animal studies, such as high heterogeneity and funnel plot asymmetry. Despite subgroup analysis on $H_2S$ donor type, species, and administration methods, the heterogeneity is not significantly reduced in the current study, probably due to potential intra-study bias, inadequate experimental design, sampling error, and other factors. These limitations may have weakened the generalizability of the study findings and increased the risk of translating the preclinical experiments into clinical studies.

## CONCLUSIONS

$H_2S$ exhibits significant nephroprotective effects on animal experimental AKI models, making it a promising candidate for the treatment of AKI induced by various causes. The beneficial effects are likely attributed to their ability to reduce oxidative stress, inflammation, and possible cell deaths. The broad applications of $H_2S$ on cis-AKI animal models seen in the current analysis, signify its potential as a game-changer in renal medicine. The consistent and positive results encourage further exploration into the mechanisms of $H_2S$-mediated renal protection. Well-designed preclinical investigations for $H_2S$ as a drug to prevent cisplatin-induced nephrotoxicity are necessary. More attention should be paid to the beneficial effects of sustained-released $H_2S$ donors and extracts from natural sources in the prevention and treatment of cis-AKI in the future.

### Funding
The authors received no funding for this work. The National Natural Science Foundation of China (81970605) supported the APC for this article. The funders had no role in study design, data collection and analysis, decision to publish, or preparation of the manuscript.

### Grant Disclosures
The following grant information was disclosed by the authors:
The National Natural Science Foundation of China: 81970605.

### Competing Interests
The authors declare that they have no competing interests.

### Author Contributions
- Zhenyuan Han conceived and designed the experiments, performed the experiments, analyzed the data, prepared figures and/or tables, authored or reviewed drafts of the article, and approved the final draft.
- Tianyu Deng conceived and designed the experiments, performed the experiments, analyzed the data, prepared figures and/or tables, and approved the final draft.

- Dechao Yan conceived and designed the experiments, performed the experiments, analyzed the data, prepared figures and/or tables, and approved the final draft.
- Yutao Jia performed the experiments, analyzed the data, prepared figures and/or tables, software, and approved the final draft.
- Jing Tang conceived and designed the experiments, prepared figures and/or tables, authored or reviewed drafts of the article, supervision, and approved the final draft.
- Xiaoyan Wang conceived and designed the experiments, authored or reviewed drafts of the article, supervision, Project administration, and approved the final draft.

## Data Availability

This is a systematic review/meta-analysis.

## Supplemental Information

Supplemental information for this article can be found online at http://dx.doi.org/10.7717/peerj.19481#supplemental-information.

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
