# Peer review of "Hydrogen sulfide protects against cisplatin-induced experimental nephrotoxicity in animal models: a systematic review and meta-analysis"

_PeerJ, doi:10.7717/peerj.19481_

## Round 0.1 · original submission · Major Revisions

Dear authors, I ask you to carefully correct the manuscript in accordance with the reviewers' fundamental comments.

Reviewer 1 ·

Basic reporting

Dr. Han et al. have written an excellent review article. However, the Cisplatin treatment impaired hydrogen sulfide production in the kidney, and because of kidney damage, this part should be added to the review with enough evidence.

In the discussion section, lines #266, action, 294, # compound, and # 303 applied; all these words need to be corrected.

Experimental design

NA

Validity of the findings

Yes

Reviewer 2 ·

Basic reporting

Can you provide additional justification for the chosen methodologies and explain whether alternative approaches were considered? How do you ensure the reliability and reproducibility of the results? Are there any potential limitations in your data collection or analysis?

Experimental design

NO

Validity of the findings

NO

Additional comments

NO

Reviewer 3 ·

Basic reporting

This article explored the therapeutic potential of H2S for cisplatin-induced acute kidney injury through meta-analysis. Although the topic is novel, there aren’t many studies in the meta-analysis, and the methodologies used for animal experiments in various studies varies significantly.

Experimental design

1. Modeling information should also be provided in Table 1, such as the concentration, administration method and intervention duration of cisplatin.
2. In Figure 5, the authors compared the effects of H2S treatment on SOD, MDA, GSH, NF-κB, TNF-α, and cleaved-caspase3 levels. Were these assays performed using the same experimental methods (e.g., ELISA, WB, IHC, IF, or RT-PCR) in different articles? Results obtained using different detection methods cannot be combined and compared.
3. Figure 6 compares the therapeutic effects of H2S on renal tubular injury. How is renal tubular injury evaluated here? Pathological score or renal tubular injury markers?
4. It is recommended to use quality assessment tools, such as the Cochrane risk bias assessment tool, STAIR checklist, CAMARADES checklist, etc., to screen some low-quality literature.

Validity of the findings

There are too many variables and the results lack credibility.

Reviewer 4 ·

Basic reporting

The authors do a good job at doing meta-analysis on published animal studies to validate H2S efficacy in cis-AKI. They were quite rigorous with the statistical testing, and address many concerns regarding the bias's seen with these kinds of meta-studies. This is of significance for future clinical investigations, and promoting H2S as a new approach for treating cis- AKI

The professional english looks OK mostly, just the author need to re-read to fix the spacings between brackets and the words preceding it.

Experimental design

Looks good. Very thorough with stats.
Some comments:
3.1- what does this sentence mean? increase levels? Please clarify "Eight exogenous hydrogen sulfide donors were used to increase levels in living organisms, divided into synthetic compounds (ADT-OH, NaHS, Na2S4, and GYY4137) and natural plant extracts (Allicin, DAS, DATS, and DADS). "
Please also explain why this methodology of study selection was used for the meta analysis, and give references.

3.2- Why did the authors not only use studies for meta analysis that had low bias. Do we get similar results?

Validity of the findings

No comments. All look good.

Additional comments

Overall, I would urge the authors to make "better looking figures".
Please also have a figure on Distribution of species, administration routes and duration across studies. Another advise would be having a heat map figure of largest effect sizes obtained .

---

## Round 0.2 · Minor Revisions

Dear Dr. Han, I ask you to format the figures and tables according to the journal's requirements. Currently, the references to literature in the text, the figures, the tables, and the list of references contain numerous deviations from the journal's publishing standards. I hope that the new version of the article you sent can be approved for publication.

Reviewer 1 ·

Basic reporting

The manuscript is well-written and includes adequate literature references. It is also structured professionally, featuring tables, figures, research hypotheses, and conclusions.The manuscript is well-written and includes adequate literature references. It is also structured professionally, featuring tables, figures, research hypotheses, and conclusions.written in a professional manner and includes sufficient literature references. It is also professionally structured, with tables and figures, research hypotheses, and conclusions.

Experimental design

Dr. Han and colleagues have written a comprehensive review. The authors have included detailed information in each section, supported by references and statistical analyses.Dr. Han et al. have written the review with sufficient literature. The authors have included sufficient information in the sections, supported by references and statistical analyses.

Validity of the findings

Cisplatin is an effective chemotherapy drug that induces DNA crosslinking and causes DNA damage, ultimately leading to its anti-cancer effects. However, one of its most common side effects is kidney toxicity. Extensive research has been conducted on hydrogen sulfide (H2S) due to its significant role in protecting kidneys from acute kidney injury (AKI) in animal studies. Therefore, this meta-analysis aims to validate the efficacy of H2S in mitigating Cisplatin-induced AKI based on published animal research. This analysis is expected to provide valuable insights for future preclinical and clinical investigations focused on reducing nephrotoxicity.

Reviewer 2 ·

Basic reporting

Authors have revised the manuscript

Experimental design

After revision, authors have well written the experimental design.

Validity of the findings

Now it is ok to be published

---

## Round 0.3 · accepted · Accept

Dear Dr. Han, I congratulate you on the acceptance of this article for publication and hope that you will continue your research and publish more such high-quality articles in our journal.

Reviewer 4 ·

Basic reporting

Looks good.

Experimental design

Looks good.

Validity of the findings

N/A

Additional comments

The authors have sufficiently addressed all my comments.